# Future of Dutch NGS-Based Newborn Screening: Exploring the Technical Possibilities and Assessment of a Variant Classification Strategy

**DOI:** 10.3390/ijns10010020

**Published:** 2024-03-07

**Authors:** Gea Kiewiet, Dineke Westra, Eddy N. de Boer, Emma van Berkel, Tom G. J. Hofste, Martine van Zweeden, Ronny C. Derks, Nico F. A. Leijsten, Martina H. A. Ruiterkamp-Versteeg, Bart Charbon, Lennart Johansson, Janneke Bos-Kruizinga, Inge J. Veenstra, Monique G. M. de Sain-van der Velden, Els Voorhoeve, M. Rebecca Heiner-Fokkema, Francjan van Spronsen, Birgit Sikkema-Raddatz, Marcel Nelen

**Affiliations:** 1Department of Genetics, University Medical Center Groningen, University of Groningen, 9713 GZ Groningen, The Netherlands; e.n.de.boer@umcg.nl (E.N.d.B.); b.charbon@umcg.nl (B.C.); l.johansson@umcg.nl (L.J.); j.bos-kruizinga@umcg.nl (J.B.-K.); i.j.veenstra@umcg.nl (I.J.V.); b.sikkema01@umcg.nl (B.S.-R.); 2Department of Human Genetics, Radboud University Medical Center, 6500 HB Nijmegen, The Netherlands; dineke.westra@radboudumc.nl (D.W.); emma.vanberkel@radboudumc.nl (E.v.B.); tom.hofste@radboudumc.nl (T.G.J.H.); martine.vanzweeden@radboudumc.nl (M.v.Z.); ronny.derks@radboudumc.nl (R.C.D.); nico.leijsten@radboudumc.nl (N.F.A.L.); martina.ruiterkamp-versteeg@radboudumc.nl (M.H.A.R.-V.); m.r.nelen-2@umcutrecht.nl (M.N.); 3Section Metabolic Diagnostics, Department of Genetics, University Medical Center Utrecht, 3584 CX Utrecht, The Netherlands; m.g.desain@umcutrecht.nl; 4Centre for Health Protection, National Institute for Public Health and the Environment, 3720 BA Bilthoven, The Netherlands; els.voorhoeve@rivm.nl; 5Department of Laboratory Medicine, University Medical Center Groningen, University of Groningen, 9713 GZ Groningen, The Netherlands; m.r.heiner@umcg.nl; 6Division of Metabolic Diseases, Beatrix Children’s Hospital, University Medical Center Groningen, University of Groningen, 9713 GZ Groningen, The Netherlands; f.j.van.spronsen@umcg.nl; 7Department of Genetics, University Medical Center Utrecht, 3584 CX Utrecht, The Netherlands

**Keywords:** newborn screening, next-generation sequencing, inherited metabolic disorder, dried blood spots

## Abstract

In this study, we compare next-generation sequencing (NGS) approaches (targeted panel (tNGS), whole exome sequencing (WES), and whole genome sequencing (WGS)) for application in newborn screening (NBS). DNA was extracted from dried blood spots (DBS) from 50 patients with genetically confirmed inherited metabolic disorders (IMDs) and 50 control samples. One hundred IMD-related genes were analyzed. Two data-filtering strategies were applied: one to detect only (likely) pathogenic ((L)P) variants, and one to detect (L)P variants in combination with variants of unknown significance (VUS). The variants were filtered and interpreted, defining true/false positives (TP/FP) and true/false negatives (TN/FN). The variant filtering strategies were assessed in a background cohort (BC) of 4833 individuals. Reliable results were obtained within 5 days. TP results (47 patient samples) for tNGS, WES, and WGS results were 33, 31, and 30, respectively, using the (L)P filtering, and 40, 40, and 38, respectively, when including VUS. FN results were 11, 13, and 14, respectively, excluding VUS, and 4, 4, and 6, when including VUS. The remaining FN were mainly samples with a homozygous VUS. All controls were TN. Three BC individuals showed a homozygous (L)P variant, all related to a variable, mild phenotype. The use of NGS-based workflows in NBS seems promising, although more knowledge of data handling, automated variant interpretation, and costs is needed before implementation.

## 1. Introduction

Many countries worldwide use newborn screening (NBS) programs for treatable disorders [1,2]. The Dutch NBS program tests for 27 disorders, including 19 inherited metabolic disorders (IMDs) with a monogenetic origin [2]. Since the introduction of NBS, there has been ongoing demand for its expansion and for a reduction in the proportion of false positives (FP) [3]. However, including new disorders in the NBS program is labor-intensive, because disorder-specific tests need to be developed, while reducing FP remains challenging [4,5,6]. Furthermore, the biochemical testing approach hampers the inclusion of disorders that lack a biochemical footprint. As IMDs are hereditary, genetic testing using next-generation sequencing (NGS) techniques can overcome these shortcomings. NGS-based genetic testing is an all-in-one assay that can easily be modified to include new disorders, and NGS is already widely used in clinical care to diagnose patients with a suspected genetic condition [7,8,9,10,11]. Several recent studies also started to explore the use of NGS-based tests in NBS [12,13].

However, while the advantages of NGS-based genetic testing are widely recognized, many challenges remain [14,15]. One of these is selection of the most suitable NGS approach. In principle, targeted sequencing of a panel of genes (tNGS), whole-exome sequencing (WES), and whole-genome sequencing (WGS) are all reliable methods for detecting genetic disorders in a high-throughput manner, and tNGS and WES have proven their usefulness in the clinical setting. At the same time, tNGS, WES, and WGS differ in their ability to detect different types of variants, their turnaround times, and their costs. Furthermore, high-throughput data analysis, filtering, and automated variant interpretation would be necessary in NBS where fast turnaround times are needed, and the majority of newborns are healthy. In this study we explored three different NGS approaches for use in NBS: tNGS, WES, and WGS performed on DNA obtained from dried blood spots (DBS). The techniques were compared with respect to technical aspects, variant detection, interpretation, and turnaround times. We also assessed the variant classification strategy we used in a background cohort to identify the potential number of FP. The outcomes of this study provide recommendations for the implementation of a large-scale NGS-based approach.

## 2. Materials and Methods

### 2.1. Experimental Setup

In total, 100 genes associated with 95 metabolic disorders were previously selected as candidates for inclusion in a NGS-based NBS (Appendix A) (Veldman et al. manuscript in preparation). Disease severity, treatability, and age of onset were taken as the main inclusion criteria. To test the performance of different NGS techniques for use in NBS, we compiled a test cohort of 100 DBS: 50 from patients with a genetically confirmed IMD and 50 from healthy controls. After DNA extraction (using 1 blood spot in total), tNGS and WES were performed on all samples. WGS was performed on the 50 DBS from patients. All analyses were carried out following standard procedures used in diagnostics (Figure 1), with both single nucleotide variants (SNVs) and copy number variants (CNVs) analyzed. After data processing and the application of a virtual panel to the WES and WGS data, variant filtering was performed. Variants were assessed and interpreted blindly, with the outcomes of the NGS approaches and previous genetic results then compared ‘post-blind’. To assess the SNV variant filtering strategy, it was retrospectively applied to WES data from a cohort of 4833 individuals (further referred to as the background cohort (BC)).

### 2.2. Sample Collection and DNA Extraction

Fifty DBS from patients with an IMD associated with one of the selected one hundred genes were kindly provided by the sample archives of the Laboratory of Metabolic Diseases of the University Medical Center Groningen (UMCG) and the Laboratory of Metabolic Diseases of the University Medical Center Utrecht. Samples were only included when no patient or parental non-consent for use for research purposes was registered in the research registries of the UMCs. The Ethical Review Board (ERB) of the UMCG has declared that this study does not fall under the Medical Research Involving Humans Act, so no further ERB approval was needed (METc 2021/611). In all patients, a genetic diagnosis that fits the aberrant metabolic results had been made in a clinical setting. Fifty DBS from randomly selected anonymous blood samples from individuals without a (known) IMD served as negative controls. Preparation of these DBS and DNA extraction from all DBS are described in Appendix A. An existing WES dataset of 4833 anonymized (METc Radboud university medical center (number 2011-188 and 2020-7142)) individuals was included to compare the SNV results of the DBS in a BC representing a large background population. These BC data were collected previously in a diagnostic setting, coming from presumed healthy individuals who were included in trio-analysis (parents of an affected child). All BC individuals were born before the first IMDs, aside from phenylketonuria, became target diseases in the Dutch NBS in 2007.

### 2.3. Preparation of Samples, Sequencing, Data Filtering, and Classification

A detailed description of sample preparation and sequencing for all NGS techniques is provided in Appendix A, as are the methods for alignment, coverage calculation and variant calling. The filtering strategies, which were applied to all data, are shown in Figure 2. To start, we applied a strict filtering strategy designed to only detect pathogenic (P) and likely pathogenic (LP) variants, according to recommendations for other genetic screening approaches [16] and in line with other studies [17]. We included filter steps that selected (1) variants conclusively described as P and LP in any of the following variant databases (either a disease-causing mutation in the Human Gene Mutation Database pro v.2022, (L)P in ClinVar23 (last update 6 October 2022), (L)P in the national database of the Dutch Society of Clinical Laboratory Genetics (December 2021 release, https://vkgl.molgeniscloud.org/ accessed on 1 May 2022), or (L)P in local databases), (2) truncating variants not described in databases (nonsense, frameshift, start/stoploss, canonical splice site; unless present downstream of the nonsense-mediated decay boundary) [18], and (3) variants with possible splicing effects (positions ± 3). Filtered variants also needed to have an allele frequency <1% (Figure 2A). The variants that remained after these filtering steps were evaluated by a clinical laboratory geneticist and classified according to the American College of Medical Genetics and Genomics (ACMG) guidelines [19], considering the mode of inheritance (MOI) of the associated disease. Afterwards, we extended the filtering strategy to obtain VUS from genes in which one P or LP variant with a variant allele frequency (VAF) of 40–60% was found in an autosomal recessive (AR) or X-linked recessive (XLR) gene (Figure 2B). All identified variants and combinations thereof were manually reviewed and, if necessary, reclassified.

### 2.4. Outcome of the Study

To compare the performance of the NGS workflows, we measured the following outcomes:**Technical performance of tNGS, WES, and WGS**. The number of failed samples, overall coverage, and specific regions not covered >20x were reported.**Outcomes of the two variant filtering strategies**. First, we measured the number of variants using the strict filtering strategy reporting only P and LP variants ((L)P) filter strategy; Figure 2B, left). We then calculated the number of true positives (TP), true negatives (TN), FP and false negatives (FN) based on the definitions described in Appendix A. Second, we tested a filtering strategy with an additional step (Figure 2B, right). When only one P or LP variant was found in an AR or XLR gene, we also reported VUS found in the same gene (extra VUS filter strategy). Here, we used the same definitions for TP/FP/FN/TN, but the samples in which one LP or P variant and a VUS were detected, were now considered positive. Carriership was defined as presence of one LP or P variant detected with a VAF of 50% in an AR gene in any sample.**Estimated turnaround time**. For a series of 96 samples, we measured the time needed to obtain results. We included the time needed for automated sample preparation, sequencing, data processing, and data analysis. For tNGS data analysis, an automated variant interpretation pipeline was used to obtain relevant variants for each sample (https://github.com/molgenis/vip, accessed on 1 May 2022). For WES and WGS, downstream processing was performed using an automated data analysis pipeline and custom-made annotation [20], with a bioinformatic filter for the selected genes.

## 3. Results

### 3.1. Technical Performance

The DNA isolation procedure yielded an average 702.45 ng DNA per sample (range 393.4–3458.4 ng). No data could be obtained due to the failure of library preparation for 3/100 samples in all 3 approaches. All the DNA samples that failed were archival DBS. The overall coverage for tNGS and WES fulfilled local diagnostic standard requirements (>20x coverage for >95% of the target regions). For WGS, >85% (38/44) of samples had >20x coverage. An overview of the genome positions that did not comply with these requirements is shown, per method, in Appendix A. The average read depths were 195 ± 74 for tNGS, 182.5 ± 36.7 for WES, and 31.4 ± 11.6 for WGS.

### 3.2. Filter Strategy Assessment

We calculated the average number of variants remaining after filtering using the strict (L)P filter strategy. On average, 2.4 variants (range 0–6) were found using tNGS, and 1.3 variants (range 0–3) were found using the WES and WGS workflow in the IMD-positive samples. In the control samples, a mean of 0.89 variants (range 0–4) were left after filtering using tNGS, and this mean was 0.17 (range 0–2) after WES-based analysis.

Using the strict (L)P filtering strategy (Figure 2A), 33 of the IMD samples were TP using the tNGS workflow, as were 31 when using WES and 30 when using WGS (Figure 2C). For WGS, 7/9 samples with low coverage (<20x) were correctly identified as TP. However, 11 (tNGS), 13 (WES), and 14 (WGS) samples were found to be FN. For both tNGS and WES, the 50 control samples were TN.

When we applied the VUS filter strategy (Figure 2B), 40 samples were TP in the tNGS and WES workflows, as were 38 in the WGS workflow (Figure 2C). Comparing these numbers with the TP numbers of the strict filtering strategy shows that the majority of FN samples were resolved using this strategy. Nonetheless, four (tNGS and WES) and six (WGS) samples were found to be FN. Table 1 provides a detailed overview of the variants detected per sample, technique, and filter strategy.

The additional use of the VUS filter strategy identified three samples (#31, #35, and #37) to be FN in all three workflows. The known variant in these samples was registered as a homozygous VUS in the databases used (Table 2). Sample #6 contained a disease-causing intronic variant, c.-149G>A in *SLC22A5*, that was only seen with tNGS, as non-coding regions (in this case the 5′UTR of the gene) were not included in the bioinformatic analysis for WES and WGS. Manual curation of the data, however, did identify the variant for WES and WGS. Variants in *CBS* (#3) and one variant in *ABCD1* (#31) were not identified in the WGS workflow because of reduced coverage (<20x) and known pseudogenes that complicated the analysis for these genes.

Both of the known CNVs in the IMD-positive samples were detected in all workflows. However, the WES and WGS data analysis regarding CNVs failed in 14 of 47 samples due to reference pool issues.

For the BC (Figure 3), we first looked for the presence of any SNV in the coding and splicing region (MANE transcript [21], if applicable) of the 100 genes, and 336 individuals (7%) did not have any SNV in any of the genes. The pathogenicity of the SNVs present in the remaining 4497 individuals were evaluated via the above-mentioned strict strategy, and no (L)P variant was identified in 3394 individuals (70%). Considering the VAF and the MOI of the genes in which (L)P variants were found, the number or zygosity of the (L)P variants fitted the MOI in 30 individuals (0.6%). After reviewing these variants (excluding deletion–insertions located in *cis*, but called as separate events, sequencing artefacts, and variants not relevant based on predictions and literature), a homozygous pathogenic variant remained in three individuals in *ACADVL*, *CPT1A*, and *MCCC2*, respectively (Appendix A). These variants are associated with a variable phenotype that is mostly asymptomatic or mild (Appendix A) and not with severe disease.

In 1073 individuals (22%), only 1 (L)P with a VAF of 40–60% was seen in an AR or XLR gene. For these individuals, the VUS filter strategy was applied, which resulted in an extra 18 individuals with 1 (L)P variant and a VUS fitting the MOI of the gene. After reviewing these variants, a combination of (L)P and relevant VUS in an AR gene was seen in four individuals (Appendix A).

### 3.3. Carriership

In the 100 DBS samples, 28 carrierships were detected (Appendix A) with a maximum of 1 carriership per sample. In total, 23 carrierships were detected with tNGS, 12 with WES and 12 with WGS. The differences in detection were mostly due to the presence of a (L)P classified variant in the local database. Carriership was detected in 21 different genes, with the most variants (n = 2) detected in the *PAH* and *SI* genes. In sample #2, with two known variants in *GCDH*, the outcome remained inconclusive after the WES and WGS workflows because one *GCDH* variant was filtered out, while a variant (carriership) in *BTD* was found.

### 3.4. Turnaround Time

The workflow and observed turnaround time of the experiments are summarized in Figure 4. The procedure from library preparation through variant interpretation was performed in 5 working days for both tNGS and WES. Automated library preparation took two days, followed by approximately 28 h of sequencing. The following 36 h were devoted to data processing, after which variants were filtered and prioritized to obtain the relevant variants for each sample. On day 5, all variants were manually assessed by an experienced technician and clinical laboratory geneticist. WGS was performed in 4 days, with the main difference being the enrichment process needed for WES and tNGS.

## 4. Discussion

In this study we demonstrate that using NGS as a first tier in NBS is a promising and realistic approach. We show that all three NGS strategies perform equally well and that results can be reported within 4 to 5 working days. Even though no variants associated with a severe phenotype were present in the BC, the experimental setup clearly demonstrated the need for robust filter and interpretation strategies.

We detected no relevant differences between tNGS and WES regarding technical performance. Only a few samples failed, all of them from archived DBS, in line with data showing that DNA from archived DBS deteriorates over time, increasing the risk of failure [22]. However, implementation of robust quality control parameters will allow early detection, limiting failure rates in sequencing. The NGS procedure can, thus, be repeated with little delay. For WGS, further testing is needed to confirm that an average WGS coverage of ~30x provides the sensitivity required in NBS programs. CNV calling for the tNGS data using the 100 DBS samples as a reference pool was successful for all samples, while CNV calling for WES and WGS was hampered by the use of a reference pool of EDTA blood samples. This shows the importance of using a DBS-specific reference pool. Carriership of one (L)P variant was found in 28% of the samples but was not considered for reporting since this is not within the scope of NBS, which is identifying treatable disorders in newborns.

Our filter strategy assessment showed that only a limited number of variants per sample needed manual interpretation for both the DBS and the BC, using a strict (L)P filter strategy. Although the number of variants that needed to be interpreted was low, this process has to be optimized, as NBS screening procedures are carried out on large scales. One important aspect for reducing the number of variants is the need for a highly NBS-focused, curated, open access variant database. This need is emphasized by the small differences in outcome that we observed between NGS workflows due to differences in curated variants in local databases and to experimental setup (with, e.g., coverage of non-coding regions in tNGS, but not in WES/WGS). The Kingsmore et al. study that evaluated the pathogenicity of almost 30,000 variants in 317 genes is a good starting point for the development of such a database [17].

In current Dutch biochemical NBS, the FP rate is relatively high, at least for a selection of disorders, with an overall positive predictive value of 43% reported for 2021 [23,24,25]. This is because some biomarkers are influenced by, e.g., (maternal) diet, medication, and maternal metabolism [24]. This not only creates unnecessary anxiety in parents, but the current FP rate also leads to costs due to follow-up of newborns, including more healthcare consumption [26]. In our study, we observed a false positive rate of 0.06% (3/4883), when combining the negative DBS and the BC and using our strict filter strategy. The identified variants in *ACADVL*, *CPT1A*, and *MCCC2* are associated with variable but mostly asymptomatic or mild phenotypes (Appendix A), and it is questionable whether they should be reported in NBS, as the severe disease form is the target disease. In a pre-implementation research setting, we suggest discussing the relevance of variants like these in an expert panel. Gained knowledge on these kind of variants can later be used to decide whether these variants should still be reported in NBS in the future. An interesting approach to further decreasing the number of FPs is to combine both biochemical and genetic techniques, usually with NGS as second tier, but also in a procedure where both NGS and biomarkers are used in a first-tier setting [27,28,29].

Reducing the number of FNs is a key goal in a genetics-first NBS. In our study, most of the cases contained one LP or P variant, as well as a VUS in the same gene. The percentage decreased to 6–9% when applying the VUS filter strategy, but these numbers are still high compared to the FN rate of current NBS. The key reason for this is the challenge of interpreting NGS data in the absence of any phenotypic information, a problem intrinsic to the screening setting. In the current study, variant classification was carried out according to current ACMG guidelines, i.e., what is known from various variant databases and the literature. This finding again emphasizes the importance of NBS-focused databases of curated variants. In addition, we need a biochemical and/or clinical evaluation of a (suspicious) VUS in a specific gene [30]. For many of the 95 IMD investigated, functional testing is possible (Arar et al., manuscript in preparation). Furthermore, the differences in sensitivity and specificity between biochemical methods and NGS highlight the need for a proper comparison between the yield of the two strategies when used as a first tier. It is possible that a combination of both methods will be optimal, at least for some disorders.

With our experiments, we demonstrated that automated, high-throughput NGS analysis of DBS is feasible. The NGS experiment, from library preparation to data interpretation, was carried out in a maximum of 5 days. In current Dutch NBS, the target is a first-tier result within 10 days after birth, with DBS sampling occurring on day 3 through day 7. With sampling at birth (DBS or buccal swab), NGS results can be obtained within the first week after birth. WGS offers both the fastest workflow and the possibility to use the complete genomic data.

However, before NGS in NBS can be implemented, several additional aspects need to be considered. One is testing capacity. Based on ~170,000 babies born annually in the Netherlands (and assuming almost 100% participation), we calculate a need to analyze >640 samples per day in a 5-day laboratory service. This requires 7 library prep runs per day (in a 96-well plate format). Sequencing would require multiple runs, depending on the strategy of choice. The most demanding sequencing strategy would be WGS (mean coverage 30x, 120 Gb per sample), which would require around ten NovaSeqX+ 25B flowcells (output 8 Tb) per day (which equals ten machines (dual flowcells)), as the runtime is 48 h. Another crucial aspect is the amount of sequencing data generated. Secure data storage capacity, as well as duration of storage and related costs for the different NGS techniques, needs further discussion [31]. Data analysis needs to be fully automated for all negative samples, in order to provide a fast result and to not require too much work from lab technicians and clinical laboratory geneticists. The current voluntary participation rate in NBS in the Netherlands is almost 100% [25]. To secure this for the future, investigation of the opinions of parents and the general population regarding storage and use of genetic data will be key for developing a widely supported plan for the implementation of NGS in NBS [31,32,33]. Thus, we emphasize that such ethical, social, and legal topics need attention in future studies, as well as studies on the cost-effectiveness of a genetics approach for NBS.

In conclusion, we show that all three NGS methods tested can be used to obtain reliable data from DBS samples within 5 days. Based on our results, data analysis strategies need to be optimized to enable a significant reduction in FN results. Assessment of the filter strategy when used in the BC resulted in a low number of FPs. Large cohort studies are needed for better variant classification and to learn more about large-scale data handling, turnaround times and costs involved. Based on this study, we feel that the use of NGS-based screening will benefit NBS, particularly for diseases with a high FP rate in current biochemical NBS. NGS implementation also comes with the flexibility to add new target diseases to the screening program, especially those without a biochemical marker.

## Figures and Tables

**Figure 1 IJNS-10-00020-f001:**
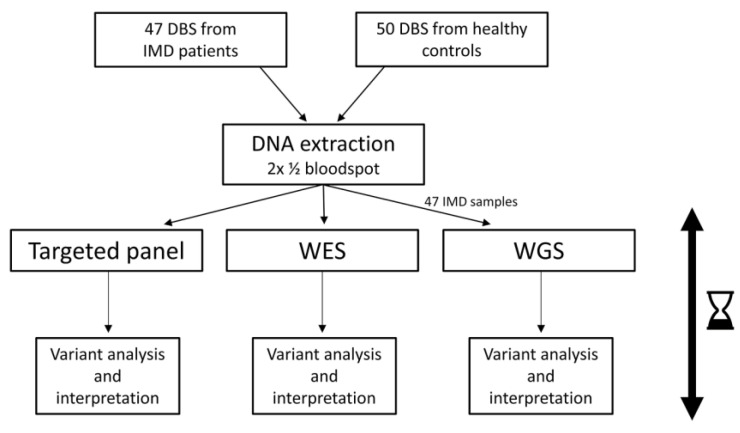
Study design. One hundred dried blood spots were analyzed using three different NGS methods, and the turnaround time of sample analysis and variant interpretation were assessed (indicated with the hourglass). DBS: dried blood spots; IMD: inherited metabolic disorder; WES: whole-exome sequencing; WGS: whole-genome sequencing.

**Figure 2 IJNS-10-00020-f002:**
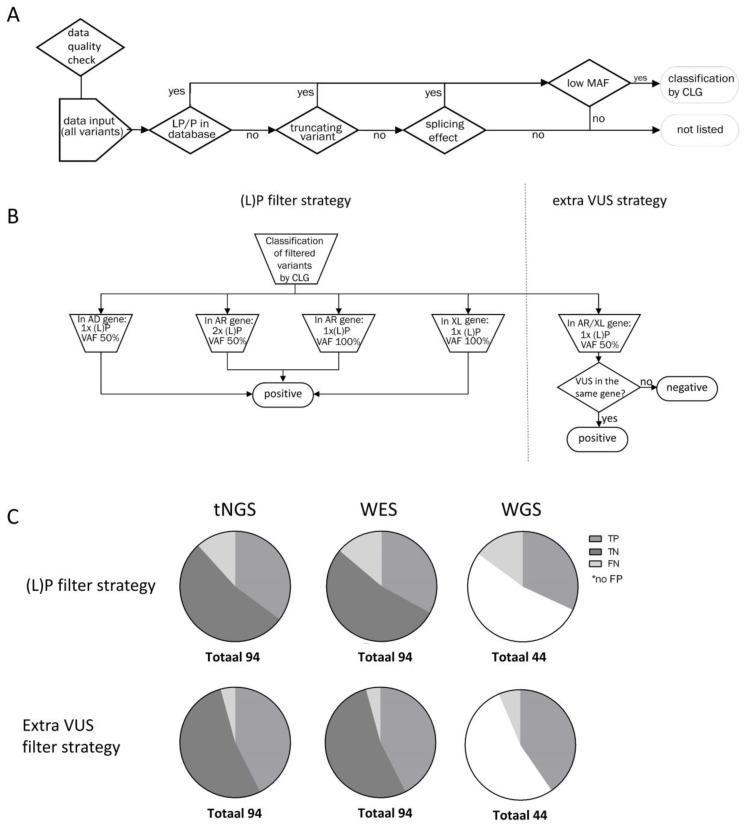
Variant filtering and outcome in dried blood spots. (**A**) Filtering steps used in the (likely) pathogenic ((L)P) filtering strategy. (**B**) (L)P filtering strategy and the additional VUS strategy. (**C**) Number of TP, TN, and FN results. AD: autosomal dominant; AR: autosomal recessive; CLG: clinical laboratory geneticist; MAF: minor allele frequency; VAF: variant allele frequency; VUS: variant of unknown significance; XL: X-linked recessive.

**Figure 3 IJNS-10-00020-f003:**
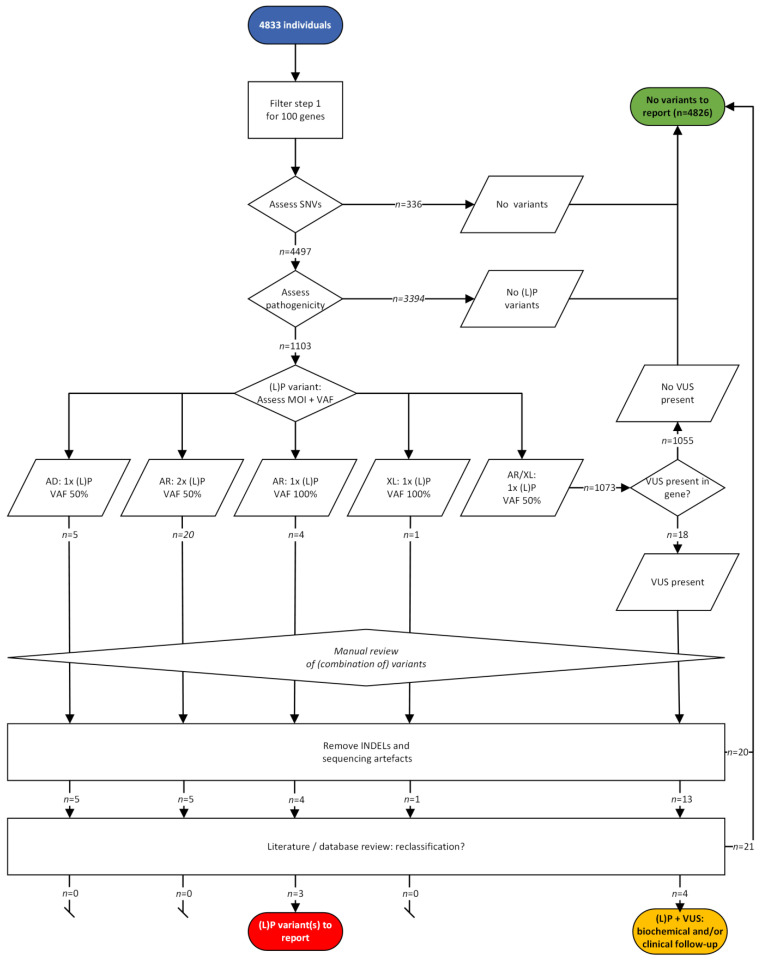
Schematic overview and outcome of the (likely) pathogenic variant filtering strategy and the extra VUS filter strategy in the background cohort. Single nucleotide variants (SNVs) present in 100 IMD genes were assessed in 4833 individuals. AD: autosomal dominant; AR: autosomal recessive; INDEL: insertion–deletion; (L)P:(likely) pathogenic; MOI: mode of inheritance; tNGS: targeted NGS; VAF: variant allele frequency; VUS: variant of unknown significance; XL: X-linked recessive.

**Figure 4 IJNS-10-00020-f004:**
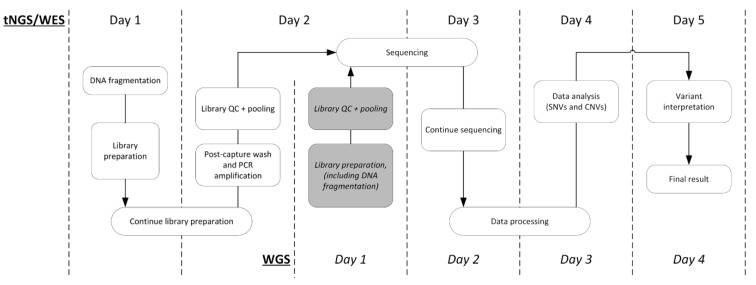
Workflows and observed turnaround times of NGS strategies including data analysis and interpretation. CNV: copy number variant; NGS: next-generation sequencing; QC: quality control; PCR: polymerase chain reaction; SNV: single nucleotide variant; tNGS: targeted next-generation sequencing; WES: whole-exome sequencing; WGS: whole-genome sequencing.

**Table 1 IJNS-10-00020-t001:** Variants detected per method per sample and considered as (L)P (“Yes”) when applying the strict (L)P filtering strategy or VUS if detected additionally using the extra VUS filtering strategy (“No *”). Hom: homozygous; het: heterozygous; hem: hemizygous; tNGS: targeted NGS; WES: whole-exome sequencing; WGS: whole-genome sequencing.

Sample	Variant	Detected?		
		tNGS	WES	WGS
1	*SLC2A2* Chr3(GRCh37):g.170716187T>C NM_000340.3:c.1771-2A>G p.?; homozygous	Yes	Yes	Yes
2	*GCDH* Chr19(GRCh37):g.13002736del NM_000159.4:c.219del p.(Tyr74fs); heterozygous	Yes	No *	No *
	*GCDH* Chr19(GRCh37):g.13004444G>A NM_000159.4:c.482G>A p.(Arg161Gln); heterozygous	Yes	Yes	Yes
3	*CBS* Chr21(GRCh37):g.44478972C>T NM_000071.3:c.1330G>A p.(Asp444Asn); heterozygous	Yes	Yes	No
	*CBS* Chr21(GRCh37):g.44484032_4484034del NM_000071.3:c.805_807del p.(Lys269del); heterozygous	Yes	Yes	No
4	*PCCB* Chr3(GRCh37):g.136035806dup NM_001178014.2:c.1050dup p.(Glu351*); heterozygous	Yes	No data	Yes
	*PCCB* Chr3(GRCh37):g.136046016_136046029delinsTAGAGCACAGGA NM_001178014.2:c.1278_1291delinsTAGAGCACAGGA p.(Gly427fs); heterozygous	Yes	No data	Yes
5	*PAH* Chr12(GRCh37):g.103234177C>T NM_000277.3:c.1315+1G>A p.?; het	Yes	Yes	Yes
	*PAH* Chr12(GRCh37):g.103288604G>T NM_000277.3:c.261C>A p.(Ser87Arg); heterozygous	Yes	Yes	Yes
6	*SLC22A5* Chr5(GRCh37):g.131705516G>A NM_003060.4:c.-149G>A p.?; heterozygous	Yes	No	No
	*SLC22A5* Chr5(GRCh37):g.131719951G>A NM_003060.4:c.610G>A p.(Gly204Ser); heterozygous	No *	No	No
7	*SLC2A1* Chr1(GRCh37):g.43395707C>G NM_006516.4:c.517-1G>C p.?; heterozygous	Yes	Yes	Yes
8	*PCCA* Chr13(GRCh37):g.100888120G>C NM_000282.4:c.625G>C p.(Ala209Pro); heterozygous	No *	No *	No *
	*PCCA* Chr13(GRCh37):g.100925458dup NM_000282.4:c.923dup p.(Leu308fs); heterozygous	Yes	Yes	Yes
9	*OAT* Chr10(GRCh37):g.126089510C>T NM_000274.4:c.1058G>A p.(Gly353Asp); heterozygous	Yes	Yes	Yes
	*OAT* Chr10(GRCh37):g.126086661del NM_000274.4:c.1171del p.(Trp391fs); heterozygous	Yes	Yes	Yes
10	*ACADM* Chr1(GRCh37):g.76198409T>C NM_000016.6:c.199T>C p.(Tyr67His); heterozygous	Yes	Yes	Yes
	*ACADM* Chr1(GRCh37):g.76226846A>G NM_000016.6:c.985A>G p.(Lys329Glu); heterozygous	Yes	Yes	Yes
11	*SLC52A3* Chr20(GRCh37):g.744576G>C NM_033409.4:c.639C>G p.(Tyr213*); heterozygous	Yes	Yes	Yes
	*SLC52A3* Chr20(GRCh37):g.744542_744544del NM_033409.4:c.678_680del p.(Leu227del); heterozygous	No *	No *	No *
12	*FAH* Chr15(GRCh37):g.80464558T>G NM_000137.4:c.674T>G p.(Ile225Ser); heterozygous	No *	No *	No *
	*FAH* Chr15(GRCh37):g.80472572G>A NM_000137.4:c.1062+5G>A p.?; heterozygous	Yes	Yes	Yes
13	*ASS1* Chr9(GRCh37):g.133352345_133352352del NM_054012.4:c.685_688+4del p.(fs232*); heterozygous	Yes	Yes	Yes
	*ASS1* Chr9(GRCh37):g.133355813G>A NM_054012.4:c.815G>A p.(Arg272His); heterozygous	Yes	Yes	Yes
14	*BCKDHB* Chr6(GRCh37):g.80982870C>T NM_183050.4:c.970C>T p.(Arg324*); homozygous	Yes	Yes	Yes
15	*GAMT* Chr19(GRCh37):g.1398988A>G NM_000156.6:c.497T>C p.(Leu166Pro); homozygous	Yes	Yes	Yes
16	*ETFB* Chr19(GRCh37):g.51848627_51848629del NM_001985.3:c.614_616delAGA p.(Lys205del); homozygous	Yes	Yes	Yes
17	*MMACHC* Chr1(GRCh37):g.45973222G>T NM_015506.3:c.276G>T p.(Glu92Asp); homozygous	Yes	Yes	Yes
18	*ACAT1* Chr11(GRCh37):g.108010834C>T NM_000019.4:c.662C>T p.(Arg208*); heterozygous	Yes	Yes	Yes
	*ACAT1* Chr1(GRCh37):g.108016927A>C NM_000019.4:c.1006-2A>C p.?; heterozygous	Yes	Yes	Yes
19	*MMUT* Chr6(GRCh37):g.49425703G>A NM_000255.4:c.454C>T p.(Arg152*); heterozygous	Yes	Yes	Yes
	*MMUT* Chr6(GRCh37):g.49425502T>A NM_00255.4:c.665A>T p.(Asn219Tyr); heterozygous	Yes	Yes	Yes
20	*OTC* ChrX(GRCh37):g.38271205C>T NM_000531.6:c.958C>T p.(Arg320*); homozygous	Yes	Yes	No data
21	*ACADVL* Chr17(GRCh37):g.7123482del NM_000018.4:c.104del p.(Pro35fs); heterozygous	Yes	Yes	Yes
	*ACADVL* Chr17(GRCh37):g.7125591T>C NM_000018.4:c.848T>C p.(Val283Ala); heterozygous	Yes	Yes	Yes
22	*SLC52A2* Chr8(GRCh37):g.145583300dup NM_001363118.2:c.148dup p.(Tyr50fs); heterozygous	Yes	Yes	Yes
	*SLC52A2* Chr8(GRCh37):g.145584264T>C NM_001363118.2:c.1016T>C p.(Leu339Pro); heterozygous	Yes	Yes	Yes
23	*MMACHC* Chr1(GRCh37):g.45973217dup NM_015506.3:c.271dup p.(Arg91fs); heterozygous	No data	No data	Yes
	*MMACHC* Chr1(GRCh37):g.45973222G>T NM_015506.3:c.276G>T p.(Glu92Asp); heterozygous	No data	No data	Yes
24	*DNAJC12* Chr10(GRCh37):g.69583144del NM_021800.3:c.85del p.(Gln29fs); heterozygous	Yes	Yes	Yes
	*DNAJC12* Chr10(GRCh37):g.69556875C>A NM_021800.3:c.596G>T p.(*199Leuext*42); heterozygous	Yes	Yes	Yes
25	*ALDH7A1* Chr5(GRCh37):g.1288206del NM_001182.5:c.1513del p.(Ala505fs); homozygous	Yes	Yes	Yes
26	*IVD* Chr15(GRCh37):g.40699855A>T NM_002225.5:c.163A>T p.(Lys55*); heterozygous	Yes	Yes	Yes
	*IVD* Chr15(GRCh37):g.40710350A>G NM_002225.3:c.1169A>G p.(Asp390Gly); heterozygous	No *	No *	No *
27	*HMGCL* Chr1(GRCh37):g.24147022C>T NM_000191.3:c.122G>A p.(Arg41Gln); homozygous	Yes	Yes	Yes
28	Chr17(GRCh37):g.3493545_3564028del; heterozygous (57 kb deletion including CTNS gene)	Yes	Yes	Yes
	*CTNS* Chr17(GRCh37):g.3543518_3543521del NM_004937.3:c.18_21del p.(Thr7fs); hemizygous	Yes	Yes	Yes
29	*G6PC* Chr17(GRCh37):g.41052972dell NM_000151.4:c.79del p.(Gln27fs); heterozygous	Yes	Yes	Yes
	*G6PC* Chr17(GRCh37):g.41063157del NM_000151.4:c.788del p.(Lys263fs); heterozygous	Yes	Yes	Yes
30	*ETFA* Chr15(GRCh37):g.76603769C>T NM_000126.4:c.-40G>A p.?: homozygous	No	No	No
31	*ABCD1* ChrX(GRCh37):g.152991164A>G NM_000033.4:c.443A>G p.(Asn148Ser); hem.	Yes	Yes	No
32	*AGL* Chr1(GRCh37):g.100316614C>T NM_000642.3:c.16C>T p.(Gln6*); heterozygous	No data	Yes	Yes
	*AGL* Chr1(GRCh37):g.100387137dup NM_000642.3:c.4529dup p.(Tyr1510*); heterozygous	No data	Yes	Yes
33	*ASL* Chr7(GRCh37):g.65551586T>C NM_000048.4:c.461T>C p.(Leu154Pro); heterozygous	Yes	No data	Yes
	*ASL* Chr7(GRCh37):g.65551738G>A NM_000048.4:c.532G>A p.(Val178Met); heterozygous	Yes	No data	Yes
34	*BCKDHA* Chr19(GRCh37):g.41916527T>A NM_000709.4:c.109-15T>A p.?; homozygous	No	No	No
35	*BTD* Chr3(GRCh37):g.15676984_15676990delinsTCC NM_000060.2:c.98_104delinsTCC p.(Cys33fs); heterozygous	Yes	Yes	Yes
	No second variant found in diagnostic setting	n.a.		
36	*CAD* Chr2(GRCh37):g.27460617C>T NM_004341.5:c.4595C>T p.(Ala1532Val); homozygous	No	No	No
37	*CPT2* Chr1(GRCh37):g.53666438C>G NM_000098.3:c.200C>G p.(Ala67Gly); heterozygous	Yes	No *	No *
	*CPT2* Chr1(GRCh37):g.53676026C>T NM_000098.3:c.680C>T p.(Pro227Leu); heterozygous	Yes	Yes	Yes
38	*CYP27A1* Chr2(GRCh37):g.219677818C>T NM_000784.4:c.1016C>T p.(Thr339Met); heterozygous	Yes	Yes	Yes
	*CYP27A1* Chr2(GRCh37):g.219678909C>T NM_000784.4:c.1183C>T p.(Arg395Cys); heterozygous	Yes	Yes	Yes
39	*FOLR1* Chr11(GRCh37):g.71906952T>C NM_016729.3:c.505T>C p.(Cys169Arg); homozygous	No	Yes	Yes
40	Chr12(GRCh37):g.1955262_22837888del; heterozygous deletion including GYS2	Yes	Yes	Yes
	*GYS2* c.495+1G>T p.?; heterozygous	Yes	Yes	Yes
41	*HADHA* Chr2(GRCh37):g.26418053C>G NM_000182.5:c.1528G>C p.(Glu510Gln); heterozygous	Yes	Yes	Yes
	*HADHA* Chr2(GRCh37):g.26414401del NM_000182.5:c.2099del p.(Gly700fs); heterozygous	Yes	Yes	Yes
42	*HMGCS2* Chr1(GRCh37):g.120307008G>A NM_005518.4:c.346C>T p.(Arg116Cys); heterozygous	No *	No	No data
	*HMGCS2* Chr1(GRCh37):g.120302538C>T NM_005518.4:c.634G>A p.(Gly212Arg); heterozygous	Yes	Yes	No data
43	*MCCC1* Chr3(GRCh37):g.18278896A>T NM_020166.5:c.639+2T>A p.?; homozygous	Yes	Yes	No data
44	*MCCC2* Chr5(GRCh37):g.70945074C>T NM_022132.5:c.1367C>T p.(Ala456Val); heterozygous	Yes	No *	No *
	*MCCC2* Chr5(GRCh37):g.70948566A>G NM_022132.5:c.1559A>G p.(Tyr520Cys); heterozygous	Yes	Yes	Yes
45	*OXCT1* Chr5(GRCh37):g.41803250C>T NM_000436.4:c.971G>A p.(Gly324Glu); homozygous	No data	Yes	Yes
46	*TH* Chr11(GRCh37):g.2189135C>T NM_199292.3:c.698G>A p.(Arg233His); heterozygous	Yes	Yes	Yes
	*TH* Chr11(GRCh37):g.2186980G>A NM_199292.3:c.1211C>T p.(Thr404Met); heterozygous	No *	No *	No *
47	*SLC2A1* Chr1(GRCh37):g.43395453T>A NM_006516.4:c.680-2A>T p.?; heterozygous	Yes	Yes	Yes

**Table 2 IJNS-10-00020-t002:** **Overview of false negative samples.** Variants not detected after applying the less strict extra VUS filter strategy are listed. tNGS: targeted NGS; WES: whole-exome sequencing; WGS: whole-genome sequencing; hom.: homozygous; AR: autosomal recessive; XL: recessive X-linked; VUS: variant of unknown significance.

tNGS			WES			WGS		
Sample	Variant (s)	Reason Variant Missed	Sample	Variant (s)	Reason Variant Missed	Sample	Variant (s)	Reason Variant Missed
			6	*SLC22A5* Chr5(GRCh37):g.131705516G>A NM_003060.4:c.-149G>A p.?; heterozygous	3′UTR variant filtered out, but present in raw data/other variant VUS	6	*SLC22A5* Chr5(GRCh37):g.131705516G>A NM_003060.4:c.-149G>A p.?; heterozygous	3′UTR variant filtered out, but present in raw data/other variant VUS
30	*ETFA* Chr15(GRCh37):g.76603769C>T NM_000126.4:c.-40G>A p.?: homozygous	hom. VUS/present in raw data	30	*ETFA* Chr15(GRCh37):g.76603769C>T NM_000126.4:c.-40G>A p.?: homozygous	hom. VUS/present in raw data	30	*ETFA* Chr15(GRCh37):g.76603769C>T NM_000126.4:c.-40G>A p.?: homozygous	hom. VUS/present in raw data
34	*BCKDHA* Chr19(GRCh37):g.41916527T>A NM_000709.4:c.109-15T>A p.?; homozygous	hom. VUS/present in raw data	34	*BCKDHA* Chr19(GRCh37):g.41916527T>A NM_000709.4:c.109-15T>A p.?; homozygous	hom. VUS/present in raw data	34	*BCKDHA* Chr19(GRCh37):g.41916527T>A NM_000709.4:c.109-15T>A p.?; homozygous	hom. VUS/present in raw data
36	*CAD* Chr2(GRCh37):g.27460617C>T NM_004341.5:c.4595C>T p.(Ala1532Val); homozygous	hom. VUS/present in raw data	36	*CAD* Chr2(GRCh37):g.27460617C>T NM_004341.5:c.4595C>T p.(Ala1532Val); homozygous	hom. VUS/present in raw data	36	*CAD* Chr2(GRCh37):g.27460617C>T NM_004341.5:c.4595C>T p.(Ala1532Val); homozygous	hom. VUS/present in raw data
39	*FOLR1* Chr11(GRCh37):g.71906952T>C NM_016729.3:c.505T>C p.(Cys169Arg); homozygous	hom. VUS/present in raw data				3	*CBS* Chr21(GRCh37):g.44478972C>T NM_000071.3:c.1330G>A p.(Asp444Asn); heterozygous *CBS* Chr21(GRCh37):g.44484032_4484034del NM_000071.3:c.805_807del p.(Lys269del); heterozygous	Low coverage and pseudogene
						31	*ABCD1* ChrX(GRCh37):g.152991164A>G NM_000033.4:c.443A>G p.(Asn148Ser); hem.	Low coverage and pseudogene

## Data Availability

The data that support the findings of this study are available from the corresponding author, G.K., upon reasonable request. Raw data files are not available due to privacy issues related to genetic data.

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
