# Peer review of "Future of Dutch NGS-Based Newborn Screening: Exploring the Technical Possibilities and Assessment of a Variant Classification Strategy"

_2409-515X, 2024, doi:10.3390/ijns10010020_

Round 1

Reviewer 1 Report

Comments and Suggestions for Authors

Very nice manuscript and thorough comparison of three NGS methods.

General variant filtration questions:

                You say the same filtration strategy was applied to all three: tNGS/WES/WGS? I’m confused why some variants are Yes for tNGS vs No* for WES/WGS then? Example: CPT2 in sample 37, GCDH in sample 2 (this is mentioned as filtered out in line 239; shouldn’t the same filtering strategy have filtered it out from tNGS?)

                For FOLR1 in sample 39, no discussion of why the homozygous variant was missed by tNGS but not WES/WGS. Was this a filtration difference? Or did tNGS not have coverage?

Lines 62-65:Furthermore, high-throughput data analysis, filtering and automated variant interpretation would be of enormous importance in NBS in which fast turnaround times are needed and the majority of newborns are healthy.”

                This might be necessary vs just enormous importance. Depending on the size of the program vs the number of samples, NGS-based NBS might simply not be implemented without automated interpretation.

Lines 174-177:

 Can you include the average number of variants with the VUS strategy for comparison?

Lines 213-214:

Explanation of why this went from 18 to 4 would be helpful. Did the VUS go LB? Did the (L)P go VUS?

Table 1:

                Adding a column for interpretation might be helpful. If transcripts are added to table E1, these could be removed from table 1 to save space.

Table E1:

                Transcript IDs would be useful here. I know the text says “MANE where available” but having the IDs documented in one place would be valuable.

Author Response

REVIEWER 1

Very nice manuscript and thorough comparison of three NGS methods.

Reply: We thank the author for the compliment.

General variant filtration questions:

                You say the same filtration strategy was applied to all three: tNGS/WES/WGS? I’m confused why some variants are Yes for tNGS vs No* for WES/WGS then? Example: CPT2 in sample 37, GCDH in sample 2 (this is mentioned as filtered out in line 239; shouldn’t the same filtering strategy have filtered it out from tNGS?)

Reply:The majority of differences in reported variants, especially between tNGS and WES are due to differences in local databases. The variant in GCDH was present in raw data obtained with all techniques, but was not detected in WES and WGS because it was included in the local database (Nijmegen) as a VUS, while it was not in the database of Groningen, where the tNGS was performed. Therefore, based on its truncating nature it was filtered out and reported as (L)P variant in this setting. When we further applied the extra VUS strategy, the variant would also be reported in the WES and WGS, as mentioned by the * in table 1.

                For FOLR1 in sample 39, no discussion of why the homozygous variant was missed by tNGS but not WES/WGS. Was this a filtration difference? Or did tNGS not have coverage?

Reply:This variant was present in the raw data, but not reported because this missense variant was considered to be a VUS (homozygous). It was not listed as (L)P in the local variant database used for tNGS (Groningen), but was in the local database used for WES and WGS (Nijmegen). This explains the difference between tNGS, WES and WGS in table 1. We have now added the information that the variant was present in the raw data in table 2.

Lines 62-65: “Furthermore, high-throughput data analysis, filtering and automated variant interpretation would be of enormous importance in NBS in which fast turnaround times are needed and the majority of newborns are healthy.”

                This might be necessary vs just enormous importance. Depending on the size of the program vs the number of samples, NGS-based NBS might simply not be implemented without automated interpretation.

Reply: We fully agree with the reviewer on this important conclusion. We have now adjusted the sentence on lines 63-66 accordingly. Manual inspection and (re-)classification of variants by hand remaining after a first automated filtering will always be needed, as depicted by the reduction of 30 potentially positive individuals to only 3 remaining after manual revision of the pathogenicity.

Lines 174-177:

 Can you include the average number of variants with the VUS strategy for comparison?

Reply: always applied the first strict (L)P filter strategy first. When also applying the extra VUS filter step we only check the gene in which a single (L)P variant was detected. Therefore, the number of extra variants that needed to be checked is very limited. Also, since this approach was only needed in the samples in which only one L(P) variant was found, 7, 9 and 8 for tNGS, WES and WGS respectively, we do not consider it helpful to give an overall average of the number of inspected variants when using the extra VUS strategy. Instead, we have now added a sentence on lines 188-190 to further clarify the ‘yield’ of additional TP samples when using this extra VUS strategy.

Lines 213-214:

Explanation of why this went from 18 to 4 would be helpful. Did the VUS go LB? Did the (L)P go VUS?

Reply: For both the strict filter strategy and the VUS strategy, we first performed a visual inspection of the data (BAM/CRAM-files) for the remaining variants. In this way we could exclude INDELs, which are often called as two or even more separate events and sequencing artefacts. For the strict filtering listed (L)P delins variant remained; no VUS were present in these individuals and 6 individuals in the VUS strategy. The variants in the remaining individuals (15 and 12, respectively) were manually reviewed in prediction programs, literature and databases. For instance, truncating variants downstream of the nonsense-mediated decay (NMD) boundary were reclassified as VUS instead of P, in-frame deletions or insertions in a protein domain were reclassified as VUS and in silico splicing effects were checked if no functional evidence was present in literature. Other functional evidence in literature, but also age-of-onset and severity of disease were considered as well. In this way, only 3 individuals remained with a homozygous (L)P variant and 4 individuals with a combination of (L)P + VUS. We have now briefly clarified this further in lines 211-213 of the manuscript.

Table 1:

                Adding a column for interpretation might be helpful. If transcripts are added to table E1, these could be removed from table 1 to save space.

Reply:The used transcript is a relevant part of genetic nomenclature of identified variants (HGVS nomenclature standards for the description of DNA, RNA, and protein sequence variants as described in PMID 26931183 and on hgvs-nomenclature.org) and as geneticists, we prefer to leave these in the variant description.
If the variant was considered (L)P in the strict filtering, it was detected (YES in column “Detected” for the respective technique); if it was considered VUS, it has a asterisks behind No in this column. We have added some extra explanation in the Table heading that might clarify this further.

Table E1:

                Transcript IDs would be useful here. I know the text says “MANE where available” but having the IDs documented in one place would be valuable.

Reply: The targeted gene panel consisted of all coding and non-coding exons and UTRs for all transcripts present in the RefSeq and Ensembl databases for all these genes, together with some specific intronic regions that were considered clinically relevant (as described in the supplementary file). For exome and genome sequencing, the bioinformatic annotation was done for all variants located within the genomic locations as depicted in Table E1, also irrespective of transcript IDs (also described in ref. 6 of the supplementals). Therefore, it is not possible to give transcript IDs in Table E1.The respective transcripts used for the variant nomenclature are all given in the different tables in both the manuscript and the supplements. If a MANE transcript was available, this was chosen for the variant nomenclature.

Reviewer 2 Report

Comments and Suggestions for Authors

The authors compared the performance of targeted gene panel, WES and WGS for 50 IEM patients with FN 11,13 and 14, respectively, excluding VUS, and 4,4 and 6, when including VUS. The data is important and adds values to this evolving field.

1. Can the author mention how much DBS samples are required for DNA extraction enough for tNGS, WES and WGS? 2 x 1/2 punches of 3.2mm? or other size. With the expansion of NBS conditions, sample quantity on the DBS cards can be a limitation.

2. 2.3 for filtering LP/P, pls state if the variant classification is conflicting, e.g. one LP and one VUS in the databases, would this be included or excluded?

3. 2.3 variants with possible splicing effects (positions+-3), it is better using 1,2,5 in the position which are more important in the splicing regions, or use +/- 5. The authors can discuss this point or retry with the filtering to see any difference.

4. the figure 2, words are not so clear, just use bigger fonts

5. Can the authors also provide the gene coverage of each gene in the target list comparing tNGS, WES and WGS? is it 100% coverage of each gene coding exons with 50 bp flanking region? maybe list in a table supplementary.

6. Good to know that CNV analysis is included. 

7. Please discuss how to handle one VUS, whether this would lead to reporting of many carrier status and or it can be treated as negative when one VUS is fine after full gene coverage and also negative CNV analysis. Otherwise, this would create quite a lot of workload to confirmation steps. 

8. Can the authors suggest or forecast how the sensitivity, specificity, FP and FN would be if their target gene panel can achieve 100% coverage of each gene? This is actually possible nowadays with hybridisation approach. 

9. Are there any TP cases with only one LP/P found but very distinctive biochemical abnormalities with/without functional enzyme confirmation? Can the authors discuss the limitation or concern in using first tier NGS approach handling such cases? or other screening strategies can be suggested with biochemical and/or genetic? 

10. It is remarkable to have NGS with data interpretation done with 5 days. However, can the authors elaborate on the additional manpower/resources required when it is really applied to NBS with much higher workload, say 100 - 200 samples or more a day within such TAT requirement?

Comments on the Quality of English Language

good

Author Response

REVIEWER 2

The authors compared the performance of targeted gene panel, WES and WGS for 50 IEM patients with FN 11,13 and 14, respectively, excluding VUS, and 4,4 and 6, when including VUS. The data is important and adds values to this evolving field.

Reply: We thank the author for the positive remark and we are happy that the added value of our work is recognized.

  1. Can the author mention how much DBS samples are required for DNA extraction enough for tNGS, WES and WGS? 2 x 1/2 punches of 3.2mm? or other size. With the expansion of NBS conditions, sample quantity on the DBS cards can be a limitation.

Reply: In this study, in which we performed three different techniques, tNGS, WES and WGS, on all samples, we used one blood spot in total to extract enough DNA. Punches were taken using a 6mm manual puncher, and DNA was extracted in two runs using half a blood spot each. This information was included in the supplementary file, and is now very briefly included in the main manuscript (line 80). The used extraction method yielded on average 700 ng of DNA. In an NBS setting, when only one NGS technique needs to be performed using 100 ng of DNA, we believe that extracting DNA from half a blood spot or even less would provide enough material. As discussed in line 326, (non-invasive) sampling at birth of i.e. buccal swap could bypass the relatively low DNA yield  from DBS.

  1. 2.3 for filtering LP/P, pls state if the variant classification is conflicting, e.g. one LP and one VUS in the databases, would this be included or excluded?

Reply: In case of conflicting classification in a database (which could be the case for example in ClinVar), a variant was not directly considered as (likely) pathogenic. Instead, it would continue to the next filter steps in our filter tree, where we select truncating or clear splicing variants with a low frequency for further assessment. In this way, we believe that we will filter out all possibly relevant variants at the end. We have schematically shown this process in figure 2A. We have now adjusted our sentence on line 121 in the main text to further explain and clarify this detail of our filter strategy.

  1. 2.3 variants with possible splicing effects (positions+-3), it is better using 1,2,5 in the position which are more important in the splicing regions, or use +/- 5. The authors can discuss this point or retry with the filtering to see any difference.

Reply: We included splicing effects at +/-3 positions to make sure we identify all variants at the +1 and +2 positions, of which we can be relatively sure to have an effect on splicing. In the ACMG/AMPO guideline, only variants in the canonical splice site (+/-2) are considered as null variants (PVS1 criterium) and are therefore considered as (L)P variants. In this experiment, we did not include any samples with variants in +3, +4 or +5, so we do not expect that changing our filter settings would have an effect on our TN outcome. However, we agree with the reviewer that it would be interesting in a follow-up study, in which new and more samples will be tested, to investigate the effect of adjusting this setting.

  1. the figure 2, words are not so clear, just use bigger fonts.

Reply: We agree with the reviewer that the text in this figure is too small. We have adjusted the figure accordingly and provided a new figure.

  1. Can the authors also provide the gene coverage of each gene in the target list comparing tNGS, WES and WGS? is it 100% coverage of each gene coding exons with 50 bp flanking region? maybe list in a table supplementary.

Reply: For this question we would like to refer the reviewer to supplementary file 2. Here we have listed the positions of the gene panel which did not fullfill the coverage criteria (>20x coverage for >95% of the target regions), for each technique separately. We hope this answers the reviewers question.  

  1. Good to know that CNV analysis is included. 

Reply: We are happy that the reviewer appreciated the complete workflow that we used.

  1. Please discuss how to handle one VUS, whether this would lead to reporting of many carrier status and or it can be treated as negative when one VUS is fine after full gene coverage and also negative CNV analysis. Otherwise, this would create quite a lot of workload to confirmation steps.

Reply: The aim of NBS is identifying affected newborns, which in most cases means the presence of two variants in a recessive genes, a heterozygote variant in a dominant gene or hemizygote variant in an X-linked gene. When these variants are present in a sample, we consider it positive, as shown in figure 2B. In all other cases, we believe the result should be communicated as being negative (indeed assuming sufficient coverage and negative CNV analysis). It is not desirable to report carriership of an (L)P or even a VUS in this setting, since this is not in line with the aim of NBS. We showed in the manuscript that carriership of an (L)P variant was detected in 28% of the tested samples, when also including VUS this number would be even higher. We agree with the reviewer that following up all these variants would not be feasible in terms of speed and workload, and we believe it is not needed when using a reliable genetic test. We have now added a comment on carriership in the discussion of the manuscript (lines 278-280) to address our opinion on reporting carriership.

  1. Can the authors suggest or forecast how the sensitivity, specificity, FP and FN would be if their target gene panel can achieve 100% coverage of each gene? This is actually possible nowadays with hybridisation approach. 

Reply: In this manuscript we showed that the main issue is the number of false negatives, we did not obtain any false positives. We also showed that the main reason for the false negatives we found was not a coverage issue (all variants were found back in the BAM files), but the presence of a homozygous VUS (table 2). This variant is probably linked to the disorder, but at the moment there is not enough evidence available to classify these variants. Therefore, we believe that the development of large international databases are most important to prevent as many false negative outcomes as possible

  1. Are there any TP cases with only one LP/P found but very distinctive biochemical abnormalities with/without functional enzyme confirmation? Can the authors discuss the limitation or concern in using first tier NGS approach handling such cases? or other screening strategies can be suggested with biochemical and/or genetic? 

Reply: Because we used anonymized samples, it was not possible to compare our genetic findings with the biochemical abnormalities of individual patients or controls. However, we believe that the reviewer addresses an important question. It is known, from our study but also from other literature, that NGS methods have a relatively higher number of false negative outcomes compared to biochemical methods, mostly due to variants with unknown function (as we also show in table 2). Biochemical methods in turn have a higher number of false positives, although differences between disorders need to be considered for both techniques. These concerns and issues were addressed in the discussion on lines 317-321: ‘For many of the 95 investigated IMD, functional testing is possible (Arar et al., manuscript in preparation). Furthermore, the differences in sensitivity and specificity between biochemical methods and NGS highlight the need for a proper comparison between the yield of the two strategies as a first tier. It is possible that a combination of both methods will turn out to be optimal, at least for some disorders.’ We believe that we cannot recommend a specific strategy based on our results in this manuscript, but that further studies comparing and combining both NGS and biochemical methods are needed.

  1. It is remarkable to have NGS with data interpretation done with 5 days. However, can the authors elaborate on the additional manpower/resources required when it is really applied to NBS with much higher workload, say 100 - 200 samples or more a day within such TAT requirement?

Reply: Based on the birth of 170.000 babies a year, we expect that around 640 samples need to be run and analyzed in the Netherlands each day. However, the majority of these samples will be negative (to compare, in 2022 around 480 children were referred to the hospital in the current newborn screening). To be able to handle the amount of samples, we therefore believe that we need to set-up an automated analysis system which will be able to generate a result for all negative samples. Only the samples with a positive result, or unclear result, need to be assessed by a technician and/or clinical laboratory geneticist. The exact number of samples that need further manual interpretation needs to be determined when this analysis system is developed further. We believe that aiming for 1% of all samples would be feasible. This means that around 30 samples need to be evaluated manually each week in the whole country (divided over multiple labs), which would only require a limited amount of workload. We have added a sentence on this topic in the discussion (lines 339-341) to address this issue in the manuscript.